# Biological Activities of Soy Protein Hydrolysate Conjugated with Mannose and Allulose

**DOI:** 10.3390/foods13193041

**Published:** 2024-09-25

**Authors:** Artorn Anuduang, Sakaewan Ounjaijean, Rewat Phongphisutthinant, Pornsiri Pitchakarn, Supakit Chaipoot, Sirinya Taya, Wason Parklak, Pairote Wiriyacharee, Kongsak Boonyapranai

**Affiliations:** 1Research Institute for Health Sciences, Chiang Mai University, Chiang Mai 50200, Thailand; a.anuduang@gmail.com (A.A.); sakaewan.o@cmu.ac.th (S.O.); toon.wason@gmail.com (W.P.); 2The Traditional Food Research and Development Unit, Multidisciplinary Research Institute (MDRI), Chiang Mai University, Chiang Mai 50200, Thailand; rewat.p@cmu.ac.th (R.P.); supakit.ch@cmu.ac.th (S.C.); sirinya.t@cmu.ac.th (S.T.); 3Department of Biochemistry, Faculty of Medicine, Chiang Mai University, Chiang Mai 50200, Thailand; pornsiri.p@cmu.ac.th; 4Faculty of Agro-Industry, Chiang Mai University, Chiang Mai 50200, Thailand; pairote.w@cmu.ac.th

**Keywords:** peptide conjugation, soy protein hydrolysate, SPHM, SPHA, supplementary products

## Abstract

The non-enzymatic conjugation of peptides through the Maillard reaction has gained attention as an effective method to enhance biological functions. This study focuses on two conjugate mixtures: crude soy protein hydrolysate (SPH) conjugated with mannose (SPHM) and crude soy protein hydrolysate conjugated with allulose (SPHA). These two mixtures were products of the Maillard reaction, also known as non-enzymatic glycation. In vitro experiments were conducted to evaluate the antioxidant, anti-pancreatic lipase, inhibition of Bovine Serum Albumin (BSA) denaturation, and anti-angiotensin converting enzyme (ACE) activities of these conjugated mixtures. The results indicate that conjugated mixtures significantly enhance the antioxidant potential demonstrated via the DPPH and FRAP assays. SPHA exhibits superior DPPH scavenging activity (280.87 ± 16.39 µg Trolox/mL) and FRAP value (38.91 ± 0.02 mg Trolox/mL). Additionally, both conjugate mixtures, at a concentration of 10 mg/mL, enhance the BSA denaturation properties, with SPHM showing slightly higher effectiveness compared to SPHA (19.78 ± 2.26% and 5.95 ± 3.89%, respectively). SPHA also shows an improvement in pancreatic lipase inhibition (29.43 ± 1.94%) when compared to the SPHM (23.34 ± 3.75%). Furthermore, both the conjugated mixtures and rare sugars exhibit ACE inhibitory properties on their own, effectively reducing ACE activity. Notably, the ACE inhibitory effects of the individual compounds and their conjugate mixtures (SPHM and SPHA) are comparable to those of positive control (Enalapril). In conclusion, SPHM and SPHA demonstrate a variety of bioactive properties, suggesting their potential use in functional foods or as ingredients in supplementary products.

## 1. Introduction

The conjugation of natural food components and other compounds at the molecular level is a promising alternative for making new materials or changing biomolecules to give them better physical and chemical properties. This strategic approach aims to enhance the attributes of a selected food constituent or create novel characteristics specifically tailored for specific applications [1,2]. Peptide-conjugated active compounds were classified as novel substances that demonstrate the potential to enhance the effectiveness of active substances in functional ingredients. This is achieved by utilizing the targeted delivery capabilities facilitated by peptide conjugation [3]. This innovative approach has attracted significant attention in functional food or ingredient development, as well as the potential for alternative treatments for various diseases.

The Maillard reaction (MR) or glycation is the non-enzymatic reaction of a reducing sugar with protein or peptides at amino groups. Two steps occur during the reaction. The sugar combines with a protein/peptide to create a stable Amadori and Heyn’s product early on. After that, many complicated reactions create brown, cross-linked fluorescent products in the advanced stage. Protein glycation reactions, protein functioning changes, and physiological changes have been extensively studied [4]. The duration of the MR can range from several days to weeks, depending on the type of product. For instance, the commercial production of black garlic requires maintaining temperatures between 40 and 90 °C and humidity levels between 60% and 90% [5]. The MR is involved in the humid–dry process. When the MR is applied to food products including longan fruit and shiitake mushrooms, it could enhance their antioxidative properties [6,7]. Beyond their use in functional food products, conjugated peptides have shown effectiveness in disease prevention and management, particularly by enhancing the targeted delivery of antimicrobial drugs [8]. Additionally, the combination of active compounds with oligopeptides holds promise for treating inflammatory diseases, neurological disorders, and metabolic conditions [9].

Soy protein hydrolysate (SPH) holds significant importance in food biotechnology and human health due to its nutritional values and varied functional properties for health benefits. The hydrolysate mixture, consisting of peptides of various sizes and sequences, is obtained through the hydrolysis of soybean flour or soybean protein isolate. These bioactive compounds exhibit anti-inflammatory, anticancer, antidiabetic, antihypertensive, and antioxidant effects [10,11,12]. Mannose, being a monosaccharide, plays a crucial role in cellular recognition and immune modulation [13]. When conjugated with a peptide, oligopeptide, or crude peptide substances, mannose could specifically target specific cells or tissues, especially immune cells, through interactions with mannose receptors. This targeted delivery mechanism not only enhances the effectiveness of therapeutic interventions but also facilitates immune modulation across a wide range of diseases [14]. Similarly, allulose, an epimer at the 3rd carbon of fructose, is present in extremely low concentrations relative to glucose and fructose, typically less than about 0.15% in many foods, thus earning the classification of a ‘rare’ sugar [15]. Research by Hossain et al. [16] demonstrated that allulose possesses the ability to protect the morphology and function of pancreatic islets. When conjugated with an oligopeptide, allulose can be directed to specific tissues or cell types, such as adipose tissue, where it may contribute to the regulation of glucose metabolism and energy balance. This targeted delivery approach for allulose presents a potential therapeutic strategy for managing metabolic disorders, including obesity and type 2 diabetes. 

This study builds upon our previous research, which explored the optimal conditions for the Maillard reaction between soy protein isolate and yeast extract, involving glucose and mannose as key components, successfully forming oligopeptide-conjugated sugars [16]. Additionally, our subsequent work detailed the peptide sizes and key amino acid compositions within these conjugates [17]. Extending these findings, the current research investigates the effects of glycation through the Maillard reaction, employing mannose and allulose to enhance the biological activities of soy protein hydrolysate (SPH). The findings offer a foundational framework for the development of novel functional ingredients for human supplementation.

## 2. Materials and Methods

### 2.1. Sample Preparations

#### 2.1.1. Preparation of Soy Protein Isolate 

Soybean flour (SF) (protein content 86.35%) (Food Great Products Co., Ltd., Bangkok, Thailand) was mixed with deionized water (DI) at a ratio of 1:5 (*w*/*v*) and agitated until a homogeneous slurry was achieved. The protein content within the SF was solubilized by adjusting the pH to a range of 8–9 using 5 M NaOH (RCI Labscan, Bangkok, Thailand) and stirring for 1 h. Subsequently, the mixture underwent centrifugation (Rotanta 460 ROBOTIC, Waltham, MA, USA) at 7000× *g* for 5 min. The soluble protein present in the supernatant was then precipitated by adjusting the pH to 5.0 with 5 M HCl (RCI Labscan, Bangkok, Thailand). The resulting precipitated protein underwent desalting using a 10 kDa dialysis tube for a period of 72 h, followed by re-centrifugation and freeze drying (Christ: Alpha 1–4 LSC plus, Osterode am Harz, Germany). The soy protein isolate (SPI) obtained was defatted using hexane, and the resulting product was stored frozen at −5 °C for future use.

#### 2.1.2. Preparation of Crude Soy Protein Hydrolysate Using Alcalase

The preparation of crude soy protein hydrolysate followed the procedure outlined by Phongphisutthinant et al. [16]. Initially, 5 g of soy protein isolate (SPI) (from Section 2.1.1) was dispersed in DI at a ratio of 1:40 *w*/*v* (SPI to DI water). The resulting mixture was then combined with Alcalase enzyme (Activity > 0.75 AU/mL) (Merck, Darmstadt, Germany). at an enzyme-to-substrate ratio (E/S) of 1% (*v*/*w* protein), utilizing a gentle agitation rate of approximately 20 rpm and maintained at 50 °C for 10 h. Subsequently, the reaction was halted by subjecting the mixture to boiling water (95 ± 2 °C) in a water bath for 15 min, followed by cooling. The resulting clear supernatant was collected, subjected to freeze-drying, and stored at 5 °C as soy protein hydrolysate (SPH) (degree of hydrolysis = 72.15%) for further use.

#### 2.1.3. Preparation of SPH Conjugates with Mannose or Allulose

SPH conjugates with mannose or allulose were prepared based on the methodology outlined by Phongphisutthinant et al. [16]. Soy protein hydrolysate (SPH) was prepared at a concentration of 1% (*w*/*v*) in 100 mL of DI. This SPH solution was subsequently combined in a 1:1 (*v*/*v*) ratio with 1% (*w*/*v*) solutions of both D-(+)-Mannose (M) (purity ≥ 95.0%) (Sigma-Aldrich, Darmstadt, Germany) and D-allulose (A) (purity 100% Crystalline) (Nutricost, Vineyard, UT, USA). SPH refers to soy protein hydrolysate, M to mannose, and A to allulose. All three materials were freeze-dried and then incubated at 60°C with 75% relative humidity for 10 days. The humidity condition was controlled via a saturated NaCl solution. After the conjugation process, the soy protein hydrolysate–allulose (SPHA) complex exhibited a free amino acid content of 19.31%. In comparison, the soy protein hydrolysate–mannose (SPHM) complex demonstrated a free amino acid content of 15.36%. The SPHA, SPHM, SPH*, M*, and A* were then stored at 5 °C for further analysis.

### 2.2. Antioxidant Activity Assays

#### 2.2.1. The 2,2-Diphenyl-1-picrylhydrazyl (DPPH) Free Radical Scavenging Assay

The DPPH assay protocol for the samples underwent slight modifications as described by Zhu et al. [18]. A DPPH solution with a concentration of 0.4 mM was prepared by dissolving 7.9 mg of DPPH (Sigma, Darmstadt, Germany) in 50 mL of ethanol (Merck, Darmstadt, Germany). Subsequently, 100 µL of SPH, M, A, SPHA, and SPHM solutions at concentrations of 0.1, 1, 10, and 100 mg mL^−1^ were each mixed with 100 µL of the 0.4 mM DPPH solution and incubated in the dark for 30 min at room temperature. A blank control sample was included, utilizing distilled water. Absorbance measurements of the samples were conducted at 517 nm using a microplate reader (SPECTROstar Nano, BMG LABTECH; Ortenberg, Germany). The experiments were conducted in triplicate independent experiments. The results were calculated and presented as Trolox concentration values. 

#### 2.2.2. The 2,2′-Azino-bis 3-Ethylbenzothiazoline-6-sulfonic Acid (ABTS) Assay

The ABTS assay was conducted following a method slightly modified from Anuduang et al. [19] To prepare the stock solution, 7 mM of 2,2-azinobis-3-ethyl-benzothiazoline-6-sulfonic acid (ABTS) (Sigma, Darmstadt, Germany) reagent was combined with 2.45 mM of potassium persulfate (K_8_S_2_O_8_) (Merck, Darmstadt, Germany) at a 1:1 ratio. This mixture was left at ambient temperature for approximately 12 h. Following incubation, the ABTS stock solution was prepared by diluting the mixture with DI until the absorbance at 734 nm reached 0.70 ± 0.05, as measured using a microplate reader. Subsequently, 10 µL of SPH, M, A, SPHA, and SPHM solutions at concentrations of 0.1, 1, 10, and 100 mg/mL were combined with 290 µL of the ABTS stock solution and incubated in the dark for 10 min. The absorbance of the resulting mixture was measured at 734 nm using a microplate reader (SPECTROstar Nano, BMG LABTECH; Ortenberg, Germany). The experiments were conducted in triplicate independent experiments. The results were calculated and presented as Trolox concentration values. 

#### 2.2.3. Ferric Reducing Antioxidant Power (FRAP) Assay

The ferric reducing antioxidant power (FRAP) was assessed as reported by Chaipoot et al. [20]. In brief, 10 mM 2,4,6 tripyridyl-s-triazine (TPTZ) (Sigma, Darmstadt, Germany) was mixed with 40 mM HCl (RCI Labscan, Bangkok, Thailand), 20 mM FeCl_3_ (Sigma, Darmstadt, Germany), and 300 mM acetate buffer (pH 3.6), (10:1:1 *v*/*v*/*v*) to create the freshly generated FRAP reagent. In this experiment, 180 µL of the FRAP reagent and 20 µL of the samples (SPH, M, A, SPHA, and SPHM solutions at concentrations of 0.1, 1, 10, and 100 mg/mL) were added to 96-well plates and incubated for 5 min at room temperature in the dark to complete the reaction. The absorbance of the reacting samples was then measured at 595 nm. Trolox (Sigma, Darmstadt, Germany) (1 mg/mL) was used as a positive control, whereas distilled water was used as a blank control. The experiments were conducted in triplicate independent experiments. The results were calculated and presented as Trolox concentration values.

### 2.3. Determination of Pancreatic Lipase Activity Inhibition

The anti-pancreatic lipase activity was conducted following the methodology outlined in a previous study by Chaipoot et al. [20]. Samples (SPH, M, A, SPHM, and SPHA) at 10 mg/mL were mixed with 0.05 M 4-Nitrophenyl butyrate (p-NPB) (Sigma, Darmstadt, Germany) and 100 mM phosphate buffer at pH 7.0. Subsequently, Type II Lipase from porcine pancreas was added, and the reaction mixture was pre-incubated at 37 °C for 5 min. The reaction was then allowed to proceed at 37 °C for an additional 30 min. Orlistat was used as the positive control. Absorbances at 410 nm were measured using a microplate reader SPECTROstar Nano, BMG LABTECH; Germany). The percentage of enzyme inhibitory action was calculated using the following formula:Pancreatic lipase inhibition (%) = [(A_NC_ − A_S_)/A_NC_] × 100(1)
where A_NC_ is the absorbance of the control (no inhibitor), and A_S_ is the absorbance of the samples or positive control.

### 2.4. Inhibition of BSA Denaturation

The assessment of the individual abilities of SPH, M, A, and the conjugated samples SPHM and SPHA to inhibit protein denaturation was conducted, with modifications to the procedure outlined by Qasim et al. [21]. Samples (SPH, M, A, SPHM, and SPHA) at 10 mg/mL were prepared for each sample in a phosphate buffer (0.05 M, pH 6.3). Approximately 500 µL of each sample was mixed with 2.0 mL of 10 mg/mL bovine albumin fraction V. The reaction mixture underwent initial incubation at 37 °C for 15 min, followed by subsequent heating at 70 °C for 5 min. After cooling to room temperature, the absorbance at 660 nm of the mixture was measured. Diclofenac was employed as the positive control. All tests were performed in triplicate. The percentage of inhibition of albumin denaturation was calculated according to the following equation:Inhibition of BSA denaturation (%) = [(A_0_ − A_S_)/A_0_] × 100(2)
where A_0_ is the absorbance of the control, and A_S_ is the absorbance of the samples or the positive control.

### 2.5. Determination of ACE-Inhibition 

The angiotensin-converting enzyme (ACE) inhibition of SPH, M, A, SPHA, and SPHM was determined using the colorimetric method via the ACE kit-WST (Dojindo Inc., Kumamoto, Japan). In brief, 20 µL of samples (SPH, M, A, SPHM, and SPHA) at 10 mg/mL, Enalapril (ACE inhibitor, used as the positive control at a concentration of 1 mg/mL), and DI (as a control and blank) were added to the wells of a 96-well plate. Subsequently, 20 µL of substrate buffer was added to each well, followed by the addition of 20 µL of enzyme working solution to both the Enalapril and control wells (DI was added to the blank well). All wells containing reagent solutions were incubated at 37 °C for 1 h. After incubation, 200 µL of the indicator working solution was added to each well and incubated at 37 °C for 10 min. The absorbances of the samples, positive control, control, and blank were measured at 450 nm using a microplate reader (SPECTROstar Nano, BMG LABTECH; Germany).
(3)$$\mathrm{ACE}\;\mathrm{inhibition}\;(\%)=(\frac{\mathrm{Absorbance}\;\mathrm{of}\;\mathrm{control}-\mathrm{Absorbance}\;\mathrm{of}\;\mathrm{sample}}{Absorbance\;of\;control-Absorbance\;of\;blank})\times 100$$

### 2.6. Statistical Analysis

The results are presented as the mean ± standard deviation (SD) obtained from three independent experiments. Statistical analyses were performed using one-way ANOVA with SPSS version 20. Differences in antioxidants, pancreatic lipase inhibition, inhibition of BSA denaturation, and ACE inhibition were calculated using a completely randomized design (CRD), with a significance threshold set at *p* < 0.05. Subsequently, Duncan’s multiple range test was applied to identify significant differences among the groups.

## 3. Results

### 3.1. Antioxidant Activities

The DPPH radical scavenging assays revealed that at a concentration of 100 mg/mL, crude soy protein hydrolysate conjugated with allulose (SPHA) demonstrated a significantly higher DPPH radical scavenging activity of 280.87 ± 16.39 µg Trolox/mL compared to crude soy protein hydrolysate conjugated with mannose (SPHM) (*p* < 0.05) (Figure 1A). Conversely, individual components such as crude soy protein hydrolysate (SPH), mannose (M), allulose (A), SPH*, M*, and A exhibited very low levels of DPPH radical scavenging activity. Furthermore, the results indicate a dose-dependent relationship in DPPH radical scavenging. In the context of the ABTS assay, it was observed that at concentrations of 100 mg/mL, SPHM, SPHA, and SPH, as well as at 10 mg/mL of SPHA, exhibited significantly higher ABTS radical scavenging activity (190.82 ± 0.34, 190.42 ± 0.34, 189.33 ± 0.17, and 192.11 ± 0.79 µg Trolox/mL, respectively) (*p* < 0.05) compared to SPH at concentrations of 0.1 and 1 mg/mL, SPHM at concentrations of 0.1, 1, and 10 mg/mL, and SPHA at 0.1 and 1 mg/mL. In contrast, rare sugars including mannose (M), allulose (A), and aged mannose and aged allulose exhibited very low levels of ABTS radical scavenging activity (Figure 1B). Additionally, the trend of ABTS inhibition aligns well with the dose-dependent pattern observed in DPPH radical scavenging. In terms of the FRAP assay, it was observed that SPHA at a concentration of 100 mg/mL displayed the highest FRAP value (38.91 ± 0.02 mg Trolox/mL) (*p* < 0.05) compared to other substances (SPH, SPH*, M, M* A, A*, and SPHM at concentrations of 0.1, 1, 10, and 100 mg/mL, and SPHA concentrations of 0.1, 1, and 10 mg/mL) (Figure 1C). 

In conclusion, the glycation process did not significantly enhance the antioxidant activities of individual SPH, M, or A. However, conjugates formed through glycation, such as SPHM and SPHA, demonstrated a significant increase in antioxidant activities. 

### 3.2. Pancreatic Lipase Activity Inhibition

The result showed that SPHA and A demonstrated higher pancreatic lipase activity inhibition (29.43 ± 1.94% and 35.19 ± 0.44%, respectively) (*p* < 0.05) compared to SPH, M, and SPHM. However, it is important to note that SPHA and A exhibited pancreatic lipase activity inhibition levels lower than the positive control (Orlistat) (Figure 2).

### 3.3. Inhibition of BSA Denaturation

The results revealed that at a concentration of 10 mg/mL, crude soy protein hydrolysate conjugated with mannose (SPHM) exhibited potent inhibition of BSA denaturation (19.78 ± 2.26%) compared to individual crude soy protein hydrolysate (SPH), mannose (M), and allulose (A) and SPH conjugated with allulose (SPHA) (Figure 3). However, despite the high % inhibition of BSA denaturation demonstrated by SPHM, the values remained below those of the positive control (Diclofenac).

### 3.4. Angiotensin-Converting Enzyme Inhibition

In terms of angiotensin-converting enzyme (ACE) inhibition, the results indicate that at a concentration of 10 mg/mL, crude soy protein hydrolysate (SPH) and mannose (M) exhibited significantly higher ACE inhibition (*p* < 0.05) compared to allulose (A), crude soy protein hydrolysate conjugated with mannose (SPHM), and crude soy protein hydrolysate conjugated with allulose (SPHA) (Figure 4). Notably, the ACE inhibition of SPH and M was comparable to that of the positive control (Enalapril).

## 4. Discussion

### 4.1. Antioxidants

Antioxidants play a critical role in preserving cellular health by counteracting detrimental free radicals and averting oxidative harm. Their primary function revolves around neutralizing free radicals, molecular entities capable of disrupting cellular stability. Antioxidants achieve this by either accepting or donating electrons, thereby stabilizing the unpaired state of free radicals and interrupting the chain of oxidation reactions [22]. In DPPH radical scavenging, the nitrogen atom’s lone electron in the DPPH reagent (a stable free radical reagent) is reduced to hydrazine by acquiring a hydrogen atom (H^+^) from antioxidants [23]. Consequently, DPPH scavenging serves to assess the proton-donating capacity of samples. 

Our findings revealed that crude soy protein hydrolysates conjugated with allulose (SPHA) and mannose (SPHM) were produced via non-enzymatic glycation, specifically the Maillard reaction, which involved the conjugation of crude soy protein hydrolysate with allulose and mannose. The major amino acids in SPH were proline, lysine, phenylalanine, histidine, and tyrosine [17]. Following the reaction, these conjugated mixtures were analyzed using size-exclusion chromatography through HPLC, as described in our previous study. The analysis showed that SPHA and SPHM exhibited molecular weights ranging from 1.0 to 10.0 kDa higher than those of the crude soy protein hydrolysate [17]. However, the conjugated mixtures likely contained various forms of conjugates, thermally modified sugars, oligosaccharides that may have formed during the process, and products of SPH degradation. 

SPHA exhibited significantly higher DPPH values (280.87 ± 16.39 µg Trolox/mL) compared to individual SPHM, SPH, SPH*, M, M*, A, and A*. This suggests that when SPH is conjugated with allulose, resulting in SPHA, there is a heightened ability to donate protons to the DPPH reagent. The increase in SPHA may be attributed to the Maillard reaction (MR), which results in the formation of advanced glycation end products (AGEs) known for their strong antioxidant properties. AGEs are capable of donating electrons to neutralize free radicals (such as DPPH) more effectively than their precursor peptides [24]. Additionally, MR may induce structural modifications in the peptides, such as enhanced hydrophobicity and the creation of new functional groups [25]. The higher DPPH value observed for SPHA is consistent with the conjugation seen between α-lactalbumin and allulose [18].

Regarding ABTS radical scavenging, the ABTS assay serves as an electron transfer-based method for assessing the antioxidant activity of both lipophilic and hydrophilic compounds [19,26]. The results indicated that individual SPH and its conjugates (SPHM and SPHA) displayed comparable, elevated ABTS values (ranging from 189.33 to 190.81 µg Trolox/mL), while individual M, M*, A, and A* did not. The probable reason for the high ABTS value is that the hydrogens at both the C-terminal and N-terminal of SPH, SPHM, and SPHA, provided by the highly reactive hydroxyl groups are more likely to exhibit electron-donating activities [24]. The end-terminal of SPH, SPHM, and SPHA may comprise histidine and/or tryptophan, in addition to lysine and valine. These amino acids can scavenge hydroxyl radicals by serving as electron donors. This activity fosters a hydrophobic microenvironment within the molecule, thereby augmenting the peptide’s antioxidant capacity [27]. This observation suggests that the conjugation of rare sugars (mannose and allulose) did not have a significant impact on the ABTS value when they were conjugated with SPH. The result contrasted with the conjugate of lactoferrin and chitosan oligosaccharide, suggesting that the glycation mechanism may enhance the ABTS value [28]. 

The FRAP assay, which assesses electron-donating antioxidants, functions by facilitating the reduction of ferric ions (Fe^3+^) to ferrous ions (Fe^2+^) [29]. Analysis of the results indicates a significant increase in the FRAP value upon the conjugation of SPH (at a concentration of 100 mg/mL) with mannose (SPHM), followed by conjugation with allulose (SPHA). The notably elevated FRAP values (25.09 ± 1.12 and 38.91 ± 0.02 µg Trolox/mL, respectively) observed in these two conjugated mixtures suggest that the structural modification of SPH into SPHM and SPHA enhances their electron-donating capabilities. This observation aligns with findings from previous research, where the conjugation process similarly resulted in an augmentation of FRAP values [30].

The observed increase in antioxidant activity of SPHA and SPHM is not solely attributed to the Maillard reaction but also the binding of mannose and allulose with phenolics and the conjugated mixture. This phenomenon aligns with findings related to dried longan pulp [4]. A stands for Alulose, and M stands for Mannose, neither of which have antioxidant activities. This finding aligns with those of other rare sugars, including D-allose, D-psicose, D-glucose, and D-fructose [31]. Our results suggest that the conjugated mixture forms of SPHM and SPHA demonstrate strong antioxidant activities as evidenced via DPPH and ABTS radical scavenging assays, as well as the FRAP assay. The conjugated mixtures exhibited higher antioxidant activities than crude SPH, showing a trend of elevated antioxidant values that is quite similar to those observed in the conjugations between chitosan and casein hydrophobic peptide, chitosan and soy protein peptides, chitosan, and corn protein peptides [32,33].

SPHM and SPHA show promise as candidate substances for natural synthesis with antioxidant capacity. Further research is warranted to explore the health benefits of SPHM and SPHA in cell cultures or animal studies, to elucidate specific mechanisms and potential human health benefits.

### 4.2. Pancreatic Lipase Activity Inhibition

Pancreatic lipase functions to hydrolyze dietary triacylglycerols into diacylglycerols and subsequently into free fatty acids [34]. Failure in the digestion of triacylglycerols results in reduced absorption of free fatty acids into the bloodstream. Orlistat, a medication, effectively inhibits pancreatic lipase activity, but its usage is associated with severe adverse effects on gastrointestinal, nervous, endocrine, and renal systems, as well as interference with the absorption and efficacy of various drugs and vitamins [35]. Consequently, exploring natural pancreatic lipase inhibition methods has garnered interest. 

The results indicate that SPHA and individual allulose (A) exhibit higher rates of pancreatic lipase inhibition (approximately 29.43–35.19%) compared to individual SPH, mannose (M), and the conjugated mixture form SPHM. This phenomenon may be explained by substances containing an arginine residue, which have demonstrated potent inhibition of pancreatic lipase [36,37]. Furthermore, individual mannose (M) and allulose (A) may contribute to their ability to inhibit pancreatic lipase activity by binding to hydrophobic sites on the enzyme [37]. Although the conjugated mixture form of SPH and A enhanced pancreatic lipase activity, the presence of crude substances resulted in lower efficacy compared to orlistat. This explains why, despite the conjugation, the peptide exhibited reduced pancreatic lipase activity. These findings are consistent with those from the natural extract of *Myristica fragrans*, which also demonstrated potent pancreatic lipase inhibition, though less effective than orlistat [38]. 

Our findings suggest that SPHA and A are promising candidates for supplementary food products, demonstrating pancreatic lipase inhibition activity, consistent with that observed in other natural extracts from *Echium vulgare* L. [34], *Rosmarinus officinalis* L. (rosemary), and *Moricandia arvensis* (L.) [39].

### 4.3. Inhibition of BSA Denaturation

The denaturation of proteins is a process recognized as a contributing factor to these physiological responses. Protein denaturation involves the alteration of proteins’ tertiary and secondary structures under external stress or exposure to compounds such as strong acids or bases, concentrated inorganic salts, organic solvents, or heat. The loss of their native conformation typically renders biological proteins ineffective in performing their intended functions. Notably, the denaturation of tissue proteins stands out as a well-documented initiator of inflammation, which serves as a protective mechanism in response to tissue damage inflicted by various stimuli including physical trauma, noxious chemicals, or microbial agents [40,41,42]. 

Our findings suggest that crude soy protein hydrolysate conjugated with mannose (SPHM) exhibits a significantly higher level of BSA denaturation inhibition (approximately 20%) compared to individual SPH, A, M, SPHA refers to soy protein hydrolysate conjugated with allulose, which is categorized as a conjugated mixture. SPHM demonstrated the ability to inhibit heat-induced denaturation in BSA. Subjecting egg and bovine serum albumin to heat caused denaturation of these proteins. Therefore, antiarthritic medications that inhibit the process of protein denaturation are beneficial. The study demonstrated that prazosin effectively prevents denaturation, indicating its potential as an antiarthritic agent [21]. The capacity to inhibit albumin denaturation may be attributed to various mechanisms, including the synergistic effects of substances found in SPH and M, similar to the water extraction of Coffea arabica [41]. SPHM shows promise in controlling the production of autoantigens, thereby inhibiting protein denaturation, as evidenced by the findings of Choi (*Piper chaba*) [42]. The conjugation of sugars and amino acids via the Maillard reaction has demonstrated the potential to mitigate oxidative stress and inflammation in interferon gamma-stimulated interleukin-8 and phorbol ester (PMA)-induced Caco-2 cells [43]. Additionally, the Maillard reaction involving the glycation of salmon protein with reducing sugars has been shown to enhance anti-inflammatory activity [44]. 

SPHM has the potential to inhibit BSA denaturation, making it a candidate for further investigation of its anti-inflammatory properties in cell line studies. 

### 4.4. Angiotensin-Converting Enzyme Inhibition

Angiotensin-I converting enzyme (ACE) (EC 3.4.15.1) falls under the class of zinc proteases, necessitating the presence of zinc and chloride ions for its activation. This enzyme plays a pivotal role in regulating blood pressure within the renin–angiotensin system. ACE operates by catalyzing the conversion of the inactive decapeptide, resulting in the release of a C-terminal oligopeptide. Specifically, in cases where Xaa is not Pro and Yaa is neither Asp nor Glu, this process leads to the formation of angiotensin II, a potent vasoconstrictor octapeptide [45]. 

Experimental findings suggest that crude soy protein hydrolysate (SPH) and mannose (M) individually exhibit significant ACE inhibition, exceeding 90%, a rate comparable to that of Enalapril, a recognized ACE inhibitor. This indicates a potential interaction between SPH, M, and ACE facilitated by hydrogen bonding, akin to the mechanism of action observed in ACE inhibitors [46]. Additionally, it is proposed that SPH’s elevated ACE inhibitory activity may stem from the presence of hydrophobic amino acids such as alanine and leucine [47]. Concerning mannose, it is suggested that its mechanism of action could involve zinc interaction [48]. However, the precise molecular pathways through which peptides and sugars induce ACE inhibition remain incompletely elucidated, though competitive inhibition against ACE is likely involved. The notable ACE inhibition demonstrated by SPH and M aligns with similar findings reported in studies on protein hydrolysates derived from Thai mature silkworm [45], cottonseed byproduct [49], Indonesian sea cucumbers [50], and Thai jasmine rice bran [51].

Although the ACE inhibition of conjugated mixture forms including SPHA and SPHM demonstrates rates of approximately 80% of ACE inhibition, which is lower than that of SPH and M, these conjugated forms exhibit enhanced bioactive activities overall, including antioxidant activity, anti-pancreatic lipase activity, and inhibition of BSA denaturation properties. Consequently, the conjugation technique of peptides with rare sugars such as mannose and allulose represents an innovative approach to augmenting bioactive activity. This method holds promise as a novel alternative for the production of functional foods, functional ingredients, or supplements aimed at improving human health in the near future.

## 5. Conclusions

The aim of this study was to examine the biological properties of crude soy protein hydrolysate-conjugated mannose and allulose. The findings confirmed that the combination forms, SPHM and SPHA, exhibited potent bioactive properties, including antioxidant activities as determined via the DPPH assay, ABTS assay, and FRAP method, as well as notable inhibition of heat-induced denaturation in BSA. Our results provide evidence that SPHM, SPHA, and their individual components—SPH, M, and A—are natural substances with potential as candidates for functional foods, possessing anti-pancreatic lipase and anti-ACE inhibition activities. The outcomes associated with SPHM and SPHA suggest their potential application in various domains, such as functional foods or as functional ingredients in supplementary food products.

## Figures and Tables

**Figure 1 foods-13-03041-f001:**
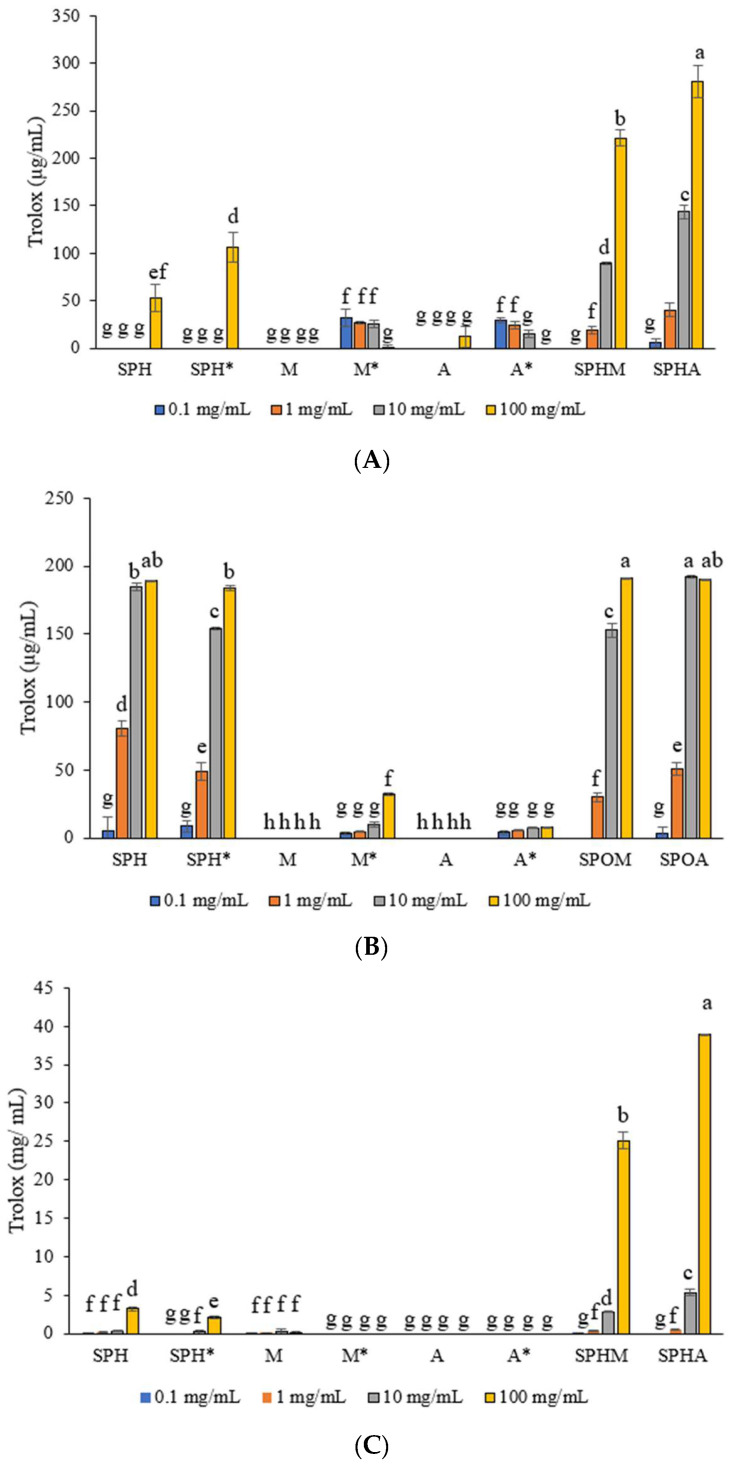
Antioxidant activities DPPH radical scavenging (**A**), ABTS radical scavenging (**B**), and FRAP essay (**C**) of crude soy protein hydrolysate: SPH, aged crude soy protein hydrolysate: SHP* mannose: M, aged mannose: M*, allulose: A, aged allulose: A*, crude soy protein hydrolysate conjugated with mannose: SPHM, and crude soy protein hydrolysate conjugated with allulose: SPHA at different concentrations, compared to Trolox. The results were expressed as the mean ± SD of triplicate measurements from independent experiments. Values of samples bearing different superscript lowercase letters are significantly different (*p* < 0.05).

**Figure 2 foods-13-03041-f002:**
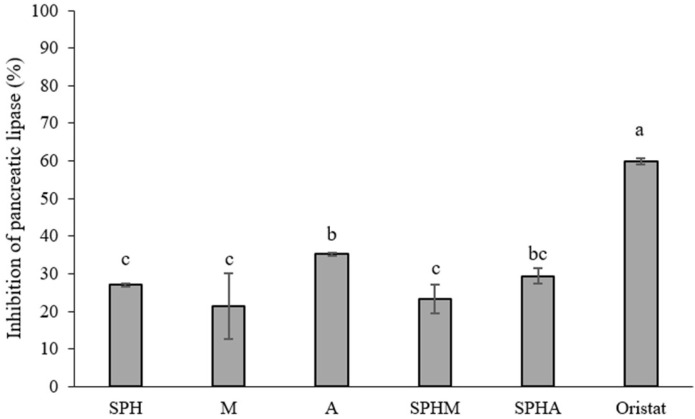
Inhibition of pancreatic lipase activity of crude soy protein hydrolysate: SPH (10 mg/mL), mannose: M (10 mg/mL), allulose: A (10 mg/mL), crude soy protein hydrolysate conjugated with mannose: SPHM (10 mg/mL), and crude soy protein hydrolysate conjugated with allulose: SPHA (10 mg/mL) compared to Orlistat (10 mg/mL). The results were expressed as the mean ± SD of triplicate measurements from independent experiments. Values of samples bearing different superscript lowercase letters are significantly different (*p* < 0.05).

**Figure 3 foods-13-03041-f003:**
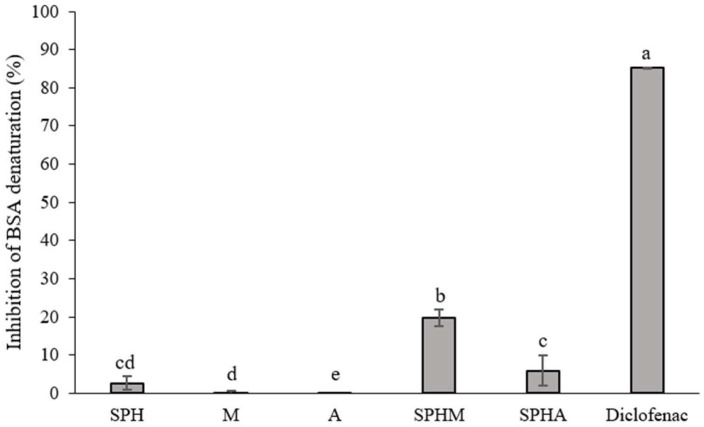
Inhibition of BSA denaturation of crude soy protein hydrolysate: SPH (10 mg/mL), mannose: M (10 mg/mL), allulose: A (10 mg/mL), soy protein hydrolysate conjugated with mannose: SPHM (10 mg/mL), and soy protein hydrolysate conjugated with allulose: SPHA (10 mg/mL) compared to Diclofenac (10 mg/mL). The results were expressed as the mean ± SD of triplicate measurements from independent experiments. Values of samples bearing different superscript lowercase letters are significantly different (*p* < 0.05).

**Figure 4 foods-13-03041-f004:**
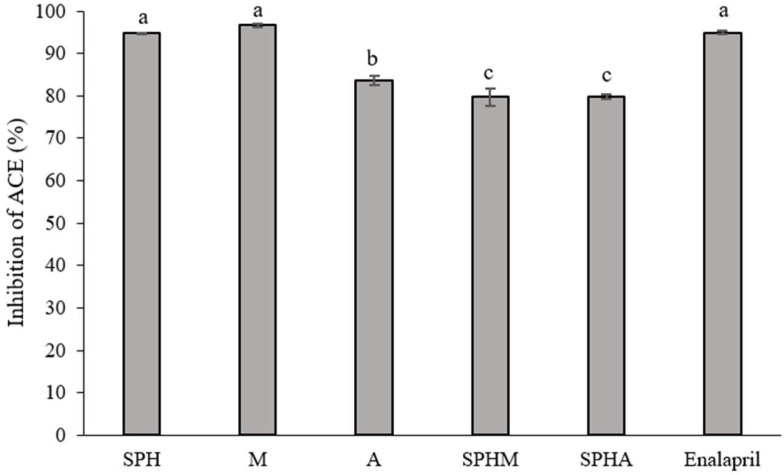
Angiotensin-converting enzyme inhibition of soy protein hydrolysate: SPH (10 mg/mL), mannose: M (10 mg/mL), allulose: A (10 mg/mL), crude soy protein hydrolysate conjugated with mannose: SPHM (10 mg/mL), and crude soy protein hydrolysate conjugated with allulose: SPHA (10 mg/mL), compared to Enalapril (1 mg/mL). The results were expressed as the mean ± SD of triplicate measurements from independent experiments. Values of samples bearing different superscript lowercase letters are significantly different (*p* < 0.05).

## Data Availability

The original contributions presented in this study are included in this article, further inquiries can be directed to the corresponding author.

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
