# Peer review of "Biological Activities of Soy Protein Hydrolysate Conjugated with Mannose and Allulose"

_foods, 2024, doi:10.3390/foods13193041_

Round 1

Reviewer 1 Report

Comments and Suggestions for Authors

This manuscript described the bioactive activities of soy protein oligopeptide conjugated with mannose and allulose. The results showed that conjugation significantly enhanced the antioxidant, anti-inflammatory, and ACE inhibitory effects. But some experiments need to be supplemented

1. The purity, structural characterization and partial physicochemical properties of soy protein, SPOM and SPOA have not been demonstrated

2. The antioxidant and anti-inflammatory activities of SPOM and SPOA were measured by biochemical methods, but not at the cellular level, so this part needs to be added.

Author Response

Dear reviewer 1 

please find the response to the comments in the attached file.

best regards

Artorn Anudnag

Reviewer 2 Report

Comments and Suggestions for Authors

The manuscript of Anuduang and colleagues is a practical, functional food study. In this study, the authors prepare soy protein Alcalase hydrolysate, perform the Maillard reaction with the monosaccharides mannose and psicose (allulose)  - btw, only the latter is a rare sugar, not both as the authors claim - and analyse antioxidant and enzyme-inhibitory action of the processed food products using spectroscopy assays in a well-plate format. From the topic and methodology perspective, the work would fit the special issue. However, several major and minor issues would restrain me from recommending this work for publishing: I suggest major revision or rejection with a possibility of re-submitting.

(i) The abstract and introduction need to be rewritten as the authors apparently used AI (ChatGPT-like) when writing, and I am not sure what the journal policy is in this regard. When reviewing, I was explicitly instructed not to use one, and therefore, I assume similar restrictions are for the manuscript authors. Even if they acknowledge AI in the methods, the text has contradictory logic that must be changed.

(ii) It is better to remove all the connotations regarding potential pharmacology-related aspects of the study objects in the introduction, discussion, and conclusions. They study processed/functional food that, apart from nutritional value, can bring supplementary health benefits but cannot be used as a chemotherapy-like drug or medication to treat diseases. Also, the methodology would need to be more appropriate for pharmacology.

(iii) The authors may like to redefine what they study (and change the narrative respectively). Strictly saying, these are not "oligopeptide/monosaccharide conjugates" but rather "products of Maillard reaction of soy protein hydrolysates with monosaccharides". The current definition of "oligopeptides" may be misleading as the term has different meanings in medicinal chemistry, pharmacology and chemical biology. Same for conjugation - in chemistry, conjugation has to be proven and compositionally defined, 

(iv) I see two general problems in all experiments: (a) proper negative controls are missing, disallowing any conclusions regarding specificity. Ideally, they would need in addition a 0 h 75% humidity 60oC  incubation mixtures ("nonconjugated", "SPO+M" and "SPO+A"), and the products of pure SPO, M and A undergoing the same processing as SPOM and SPOA (individual ingredients of the mixture incubated 10 days at the Maillard processing conditions). (b) I am unsure if the authors performed three independent experiments in each assay - the error bars are small. Presumably, they had three technical replicates (3x wells in the same plate), but not independent measurements. Otherwise, they should state if there were 3x different plates, 3x enzyme/radical batches or 3x different SPOM, SPOA, etc. preparations. Additionally, (c) in the main text and figures, the concentrations of positive controls should be present.

(v) The description of the materials needs to be included: the SF source, the sources of the sugars, and the purities need to be specified.

(vi) The antioxidant assays should uniformly be normalized to trolox.

(vii) The Protein denaturation assay cannot be claimed as "anti-inflammatory". It is just BSA denaturation interference that depends on binding to BSA. It can be speculated to be relevant to inflammation, but the latter is a much more complex in vivo process. Moreover, the assay has to be run over a series of concentrations for each denaturation perturbant and demonstrate clear concentration dependence to be speculatively relevant for the inflammation. Btw, the reference of the methods section needs to be included in the reference list. 

(viii) Advisable is to compare, in the discussion, quantitatively the effects of the studied products (e.g. antioxidant activity) with other similarly processed foods (protein extracts of different origins and or different carbohydrates processed with humid–dry heating).

Comments on the Quality of English Language

there are few spelling errorrs, but as revision requires extensive re-writting, it would make sense to check the language of the revised version. Otherwise, English is appropriate. 

Author Response

Dear reviewer 2

Thank you very much for your valuable comments. We have attached our responses to all of your comments for your review

Best regards

Artorn Anuduang 

Reviewer 3 Report

Comments and Suggestions for Authors

The research community is focusing on the production of oligopeptides and the determination of their biological properties and their application as functional food additives in view of the numerous positive effects that these compounds exhibit, therefore the topic of the manuscript is extremely interesting. After reading the paper, some questions arose that I hope the authors can answer.

Comment 1:

What is the protein content of the soy flour that you used to get the isolate?

Comment 2:

Why did you degrease the isolate with hexane after obtaining it and not protein flour at the very beginning of obtaining the isolate?

Comment 3:

When obtaining the hydrolyzate using Alcalase, did you monitor the degree of hydrolysis and do you have information on the achieved degree of hydrolysis?

Comment 4:

Do you perhaps know the molecular weight distribution of the obtained peptides and their share in the hydrolyzate?

Comment 5:

Have you perhaps tried separating the peptides and trying different combinations of peptides, polipeptides, oligopeptides with allulose and mannose?

Comment 6:

In the DPPH method, you had a positive control and a blank control in the analysis, and did you determine the share of influence of the sample itself in the solvent, in this case it is ethanol?

Suggestion:

Expand the abstract with concrete values ​​obtained in the study.

Author Response

Dear reviewer 2 

please find the response to the comments in the attached file.

best regards

Artorn Anudnag

Reviewer 4 Report

Comments and Suggestions for Authors

In this study, Anuduang et al. investigated the enhanced bioactive activities of soy protein oligopeptides conjugated with mannose and allulose, evaluating their antioxidant, anti-inflammatory, anti-pancreatic lipase, and ACE inhibitory properties. This study presents a promising approach, but there are some issues in the manuscript that need to be clarified. Here are some comments on this study:

1.        Line 45 “conjugation. [3] This”, it is recommended that authors place the reference citation within a sentence (before the period).

2.        Line 108 “deionized water (DI)”, please define the abbreviation only when it first appears.

3.        Section 2.1.3, how to verify the success of SPO conjugations preparation? We know that SPO was a mixture, whether mannose or allulose conjugated all the peptides, was it selective? It is recommended that the authors evaluated how the molecular weight distribution of the peptides changed before and after conjugation.

4.        Section 3.1, At a concentration of 10 mg/mL, SPO, M and A had no DPPH antioxidant activity, but surprisingly SPOM and SPOA had antioxidant activity, especially at the 10 mg/mL. The authors explained this reason in the discussion (lines 307-3156), but I think the authors did not explain clearly that there was some difference between the situation in the cited reference 21 and the results of the present study.

5.        Even after conjugation of M and A, pancreatic lipase activity inhibition and anti-inflammatory activity were still significantly lower compared to positive controls. So I have some doubts about the significance of conjugation process.

6.        I believe that characterization of the peptide composition of SPOM and SPOA is critical to this study, which would help to interpret the results of the study.

Author Response

Dear reviewer 3 

please find the response to the comments in the attached file.

best regards

Artorn Anudnag

Round 2

Reviewer 1 Report

Comments and Suggestions for Authors

 Accept 

Author Response

Dear Reviewer 1

Thank you so much for your valuable comments.

Best regards

Artorn Anuduang

Reviewer 2 Report

Comments and Suggestions for Authors

I am not satisfied with the fixtures in this revision. The changes are cosmetic and marginal. Moreover, I would seriously consider rejection if my suggestions are ignored or "spoken away."

Regarding my previous comments: (i) AI was not only used to check the grammar - it`s evident from the style and wordings. Even from the manuscript title ["Bioactive activities, ="biologically active activities" - is a wordy, but objectively a nonsense term], one can trace the uses of AI other than a grammar check. Anyway, the methods have to describe the use of AI explicitly (including the method of use). The abstract in this revision was not really rewritten - changes are cosmetic only: using synonyms and swapping sentence parts would not qualify as revision. Even a number of sentences remained. The introduction was marked as changed, but in reality >95% of the text remained unchanged.

(ii) This I would accept as revised

(iii) That was not addressed. The authors insist on using "oligopeptide conjugates" and add a reference to "non-enzymatic glycation". This answer does not resolve the problem: most specific conjugation methods are non-enzymatic in chemistry. The authors do not seem to understand the difference between peptide/oligopeptide/peptide mixture/protein hydrolyzate. I would insist their SPO is the latter, and this material, chemically, cannot be called "a compound" - the mixture would contain various conjugates, various non-coupled thermally modified peptides, thermally modified sugars, oligosaccharides and products of degradation thereof. 

(iva) The controls suggested are missing in this very manuscript and could not be answered as "we will measure them in the future".

(ivb) The answer does not adhere to my recommendations. The number of technical replicates is not mentioned, and the independence of the experiments is not explained.

(ivc) implemented

(v) There is no indication of stereoisomers used (complete chemically correct identification of the monosaccharides is needed)

(vi) I did not request confirmation that experiments could be performed without trolox normalization, I suggested using a uniform way for the three antioxidatiion assays - to be correct in comparing the results.

(vii) not implemented. I insist that interference with denaturation simply indicates binding and objectively cannot be described as "anti-inflammation activity". That the connotation was suggested in a not-very-reputable journal (IF=0.2) and was used several times by the other authors does not prove the correctness of the term used. It only, imho, reflects the non-robustness of reviewing/editing policies of the journals that publish such (wrong) terminology. Further: in the 2.4. the formula is wrong and  - for the same as above reasons - anti-lipase activity (2.3.) can not be called "anti-hyperlipidemia assay"; methods contain self-citations, in many cases unnecessary.

(viii) I suggested comparing quantitatively, not only mentioning the other similar products.

Author Response

(The authors gave the same response as above.)

Reviewer 4 Report

Comments and Suggestions for Authors

I thank the authors for addressing all my comments

Author Response

Dear Reviewer 4

Thank you so much for your valuable comments.

Best regards

Artorn Anuduang